# A Cost-Sensitive Diagnosis Method Based on the Operation and Maintenance Data of UAV

**Ke Zheng [1], Guozhu Jia [1], Linchao Yang [1] and Chunting Liu [2],***

[1]  School of Economics and Management, Beihang University, Beijing 100191, China;
zhengke@buaa.edu.cn (K.Z.); jiaguozhu@buaa.edu.cn (G.J.); yanglinchao@buaa.edu.cn (L.Y.)

[2]  School of Management and Economics, Beijing Institute of Technology, Beijing 100081, China

*   Correspondence: chunting@bit.edu.cn

**Abstract:** In the fault diagnosis of UAVs, extremely imbalanced data distribution and vast differences in effects of fault modes can drastically affect the application effect of a data-driven fault diagnosis model under the limitation of computing resources. At present, there is still no credible approach to determine the cost of the misdiagnosis of different fault modes that accounts for the interference of data distribution. The performance of the original cost-insensitive flight data-driven fault diagnosis models also needs to be improved. In response to this requirement, this paper proposes a two-step ensemble cost-sensitive diagnosis method based on the operation and maintenance data of UAV. According to the fault criticality from FMECA information, we defined a misdiagnosis hazard value and calculated the misdiagnosis cost. By using the misdiagnosis cost, a static cost matrix could be set to modify the diagnosis model and to evaluate the performance of the diagnosis results. A two-step ensemble cost-sensitive method based on the MetaCost framework was proposed using stratified bootstrapping, choosing LightGBM as meta-classifiers, and adjusting the ensemble form to enhance the overall performance of the diagnosis model and reduce the occupation of the computing resources while optimizing the total misdiagnosis cost. The experimental results based on the KPG component data of a large fixed-wing UAV show that the proposed cost-sensitive model can effectively reduce the total cost incurred by misdiagnosis, without putting forward excessive requirements on the computing equipment under the condition of ensuring a certain overall level of diagnosis performance.

**Keywords:** fault diagnosis; cost-sensitive learning; UAV; FMECA; MetaCost

## 1. Introduction

Unmanned aerial vehicles (UAVs), as a typical complex electromechanical system, have been widely used in the military and commercial fields but have a high fault rate. Improving the competence of fault diagnosis and ground maintenance, so as to improve the functionality and reliability of UAVs has thus become an essential research area [1–3]. With the development of Prognostics Health Management (PHM) technology, abundant onboard sensors and multisource analysis records have brought about the swift growth of operation and maintenance data of UAVs [4]. These data-driven methods, thanks to the growth of data scales, are gradually replacing the traditional Physics of Failure (PoF) methods [5,6], becoming the mainstream of fault diagnosis.

In recent years, with the rapid development of artificial intelligence technology, data-driven fault diagnosis based on machine learning models have achieved considerable progress. However, the actual data of aviation equipment are large-scale, high-dimensional, multi-class, noise-containing, and imbalanced, which poses grave challenges to generating an effective classifier. Since the loss caused by different fault modes is unequal, the cost of misdiagnosis could be great at times. As an extreme example, the Lion Air and Ethiopian Airlines crashes were caused by misdiagnoses by the Maneuvering Characteristics Augmentation System (MCAS) used in the Boeing 737 Max 8 aircraft [7]. In order to improve

the capability of the fault diagnosis model, researchers have made some inroads, focusing in particular on imbalanced data.

Due to the bias of the decision hyperplane in favor of the majority classes [8], the decent overall classification accuracy (the core indicator used to train a general classifier) might not be able to effectively diagnose more serious fault modes with lower frequencies of occurrence. Due to the great number of fault modes that exist and their great differences in frequency, the collected operation data from UAVs are greatly imbalanced. The data characteristics of the minority cases are easily outweighed by those of the majority cases, but the former often contain valuable information [9]. For industrial practitioners, this imbalanced data greatly reduces the fault detection rate and the effectiveness of a data-driven method [10]. Data imbalance significantly interferes with the generalization performance of cost-insensitive learning algorithms, which can cause the more serious faults to be ignored in fault diagnosis procedures.

To deal with this imbalanced problem, evaluation methods that focus on all classes, such as total misclassification costs [11], data-level methods, and algorithm-level methods, have been developed. Resampling, as the key to the data-level methods, tries to rebalance the data size of classes before model training with over-sampling, under-sampling, and hybrid methods. The over-sampling technique synthesizes the number of minority instances to balance the classes, whereas the under-sampling technique obtains the balanced data distribution by removing sufficient majority instances. Over-sampling methods, represented by the Synthetic Minority Over-sampling Technique (SMOTE) [12], make it difficult to fix the overfitting problem. Comparatively, under-sampling methods can lose information from partial data. These ineffaceable defects of resampling have encouraged researchers to pay increasing attention to algorithm-level methods. By adopting different types of cost-insensitive classification models, some algorithm level methods have been designed for specific data and problems [13,14], with their weak generalization ability and heavy manual work with core parameters setting becoming an issue [15]. Recently, cost-sensitive learning methods have become a popular means to solve the imbalance classification problem by considering the different misclassification costs of different classes [16,17]. In fault diagnosis, Peng et al. proposed a cost-sensitive active learning bidirectional gated recurrent unit (CSALBGRU) to reduce the effect of class imbalance [18]. An error cost function model was designed to guide the Convolutional Neural Network (CNN) parameters optimization in the direction of feature classification and was applied to the heavy-duty industrial robot system diagnosis procedure [19]. By using ensemble learning, Li et al. proposed a Dynamic Updated Ensemble (DUE) for learning imbalanced data streams with concept drift [20].

These methods evade the cost setting problem due to the difficulty of obtaining the objective and accurate misclassification cost, which means that the imbalanced problem is only related to the difference in data distribution. The effect of cost-sensitive methods still depends on the setting of the cost weight, which is also a difficulty in complex mechanical systems with unknown data distribution [15]. In fault diagnosis, the fault effect difference caused by characteristics of different fault modes cannot be ignored. Setting the misdiagnosis cost according to FMECA (Failure Mode, Effects and Criticality Analysis) [21] can eliminate the subjective effect of traditional expert scoring methods [22], and fix the scalability deficiency caused by training data deviation.

MetaCost framework [23], applicable to arbitrary cost matrices [24], is an ensemble wrapping technique that generates cost-sensitive classifiers by relabeling original training data based on minimizing loss function. Kim et al. found that using MetaCost would achieve the lowest classification cost by comparing multiple classification approaches [25]. With its enhanced robust and sensitivity, Wang and Cheng proposed a MetaCost-based combined method to process medical data [26]. Based on preferable meta-classifiers and ensemble patterns, the MetaCost method can improve the diagnosis performance in accuracy, total misdiagnosis cost, and computational resource occupation, in response to the actual demand of UAV fault diagnosis.

For identifying fault modes with more limited computational resources, this paper discusses a two-step ensemble cost-sensitive diagnosis method based on a static cost matrix. The contributions of this paper are as follows: (1) according to the fault criticality from FMECA information, we defined the misdiagnosis hazard value and calculated the misdiagnosis cost. Using the misdiagnosis cost, a static cost matrix could be set to modify the diagnosis model and to evaluate the performance of the diagnosis result. (2) A two-step ensemble cost-sensitive method based on the MetaCost Framework was proposed using the stratified bootstrapping, choosing the Light Gradient Boosting Machine (LightGBM) as a meta-classifier, and adjusting the ensemble form, to enhance the overall performance of the diagnosis model and reduce the strain on computing resources, while optimizing the total misdiagnosis cost.

The rest of this article is structured as follows: Section 2 provides a detailed description of the operation and maintenance data of the UAV discussed in this paper, the cost matrix setting method based on FMECA information, and a two-step ensemble cost-sensitive method based on the MetaCost framework. The application of and experiments with the proposed cost matrix setting method and the two-step ensemble cost-sensitive diagnosis method are described in Section 3. Section 4 presents a comparison of the methods applied to a KPG component of UAV, and the discussion. Finally, the conclusions are given in Section 5.

## 2. Materials and Methods

### 2.1. Operation and Maintenance Data of UAV

The new generation of UAVs carries numerous sensors for monitoring the operation status from self-check before takeoff to landing and shutdown. The operation data we discuss in this article is the flight data of UAVs, which is a collection of time series data points, including sensor signals and command inputs.

The characteristics of the flight data are as follows [27]:

- Large-scale. A single flight may record tens of thousands of data instances, which can be limited by a lack of computing memory.
- High-dimensional. Attained from the numerous sensors, the actual flight data of UAVs have hundreds of attributes, which can lead to the "Curse of Dimensionality".
- Multi-class. UAVs have many types of complicated fault modes, requiring the effective multi-class classification technology.
- Imbalanced. The quantity of flight data in different classes has inevitable disparities in terms of the actual operation.
- Noisy. The mission condition of UAV is so complicated that serious noise interference exists, such as the interruption of transmission or electromagnetic interference.

Other characteristics of UAV flight data include their time-sequential nature, the fact that they are recorded at fixed intervals, their inconsistent value range and accuracy, and discontinuity (outliers and frame loss). These characteristics were very important for research focusing on signal feature extraction (e.g., bearing vibration data analysis) but did not have a significant impact on our method selection.

The maintenance data we discuss in the present paper includes BIT (Build-in Test) records and FMECA information of UAVs. The new generation BIT system of UAVs, based on an expert system, PoF model, and fuzzy logic, etc., could automatically monitor most of the sensor signals or signal combinations that can cause a fault, by alerting one to or even interrupting operation instructions once the signals exceed the preset limit. Zheng utilized BIT records to label the flight data of UAVs, automatically [27]. Failure modes, effects, and criticality analysis (FMECA), a widely used methodology among the safety, reliability, and risk engineers, provide valuable information for the improvement of the safety and reliability of the process and system [28,29]. Quantitative criticality analysis (CA) could obtain the criticality value $R_i$ (to prevent confusion with the cost of misclassification) of

fault mode $F_i$ based on the probability of fault and consequences, calculated using the following equation:

$$R_i = \alpha_i \cdot \beta_i \cdot \lambda_i \cdot t \tag{1}$$

where $\alpha_i$, $\beta_i$, $\lambda_i$, and $t$ represents the fault mode ratio taken from historical database; the conditional probability of fault effect leading to identified severity classification; the basic fault rate prediction based on a specific model; and the mission phase duration, respectively [30]. As an inductive process with a "bottom-up" approach, Criticality Analysis (CA), which is meant to provide information for risk management decisions, is performed in the design and operation stages of UAV. Using this information to modify the fault diagnosis model based on flight data can effectively fuse the design knowledge and operation data together.

### 2.2. Cost Matrix Setting Based on FMECA Information

At present, the effort of imbalanced data classification is focused on the imbalance of data distribution, which fails to take full account of the essential differences between different classes of objects—for example, different fault modes. Considering that the training data used for the model cannot represent all conditions of UAV exhaustively, we chose the fault criticality value reflecting the essential difference of fault modes (from FMECA) to set the cost matrix, in order to avoid the overfitting issue led by data distribution.

For the multi-class fault diagnosis, the misdiagnosis cost $c_{ij}$ represents the classifier misdiagnoses' actual fault mode $F_j$, as predicted by fault mode $F_i$, which yields the misdiagnosis cost matrix, as shown in Table 1, which is similar to the confusion matrix.

**Table 1.** $n$-fault modes' misclassification cost matrix.

| | | Actual Fault Mode | | | |
|---|---|---|---|---|---|
| | | $F_1$ | $F_2$ | $\cdots$ | $F_n$ |
| **Predicted Fault Mode** | $F_1$ | $c_{11}$ | $c_{12}$ | $\cdots$ | $c_{1n}$ |
| | $F_2$ | $c_{21}$ | $c_{22}$ | $\cdots$ | $c_{2n}$ |
| | $\vdots$ | $\vdots$ | $\vdots$ | $\ddots$ | $\vdots$ |
| | $F_n$ | $c_{n1}$ | $c_{n2}$ | $\cdots$ | $c_{nn}$ |

For the actual fault mode $F_j$ and predicted fault mode $F_i$, we defined the misdiagnose distance of fault criticality as

$$d_{ij} = R_i - R_j \tag{2}$$

When the actual fault mode $F_j$ is more harmful than predicted fault mode $F_i$, the value of $d_{ij}$ is positive, but negative in the opposite case. Based on the misdiagnosed distance, we defined a hazard value of misdiagnosis as the following equation:

$$h_{ij} = \begin{cases} S \cdot \dfrac{d_{ij}}{\sum\limits_{i} R_i} & d_{ij} \geq 0 \\[3mm] -mS \cdot \dfrac{d_{ij}}{\sum\limits_{i} R_i} & d_{ij} < 0 \end{cases} \tag{3}$$

where $\sum\limits_{i} R_i$ indicates the sum of hazard values of all fault modes of the target equipment; $S$ represents a scaling factor to improve computing convenience (in the present paper, this is represented by $S = 100$); $m$ represents a correction coefficient between 0 and 1, for the negative value of misdiagnose distance, showing that the cost as a result of the misdiagnosis of a more serious fault compared to a minor one would be greater than the opposite. Since there is no reference to determine the value of $m$, we temporarily took $m = 0.5$ to realize the correction function and, thus, to avoid the excessive or insufficient effects.

It can be intuitively found that there is a positive correlation between the misdiagnosis cost $c_{ij}$ and the misdiagnosis hazard value $h_{ij}$, simplifying this to a linear mapping as the following:

$$c_{ij} = kh_{ij} + b \tag{4}$$

In order to simplify the computing process, the factors $k$ and $b$ can be taken as 1 and 0, respectively.

According to the above process, the misdiagnosis cost between a pair of fault modes based on FMECA information could be calculated and then used to build the misdiagnosis cost matrix.

### 2.3. Two-Step Ensemble Method Based on MetaCost Framework

### 2.3.1. MetaCost Framework

MetaCost is an ensemble framework used to convert cost-insensitive classifiers into cost-sensitive classifiers. The basic idea of MetaCost is to train multiple meta-classifiers organized by bagging first, then calculate the expected cost according to the prediction probability and misclassification cost matrix based on Bayesian theory, relabel training set based on the expected cost, and finally train an ensemble modified classifier on relabeled training data [23]. The key work of MetaCost is as follows:

Given a solid cost matrix, an instance should be classified into a class that generates the minimum expected cost. The expected cost of classifying an instance $x$ into class $C_i$ can be expressed as the following:

$$E(C_i|x) = \sum_j P(C_j|x) \cdot c_{ij} \tag{5}$$

where $P(C_j|x)$ is the probability of class $C_j$, which is the actual class of instance $x$.

Then we relabeled the instance $x$ based on the expected cost ranking of all classes as the following:

$$Label_x = \mathrm{argmin}E(C_i|x) \tag{6}$$

Subsequently, we retrained the classifier to the relabeled training set in the end. The core advantage of the MetaCost framework is its suitability for almost all kinds of data sets, thanks to the ensemble approach.

Aiming at the fault diagnosis task, the MetaCost-based cost-sensitive model can be improved in (1) the generation of training subsets for meta-classifiers; (2) the selection of meta-classifiers; and (3) the organizational form of meta-classifiers.

### 2.3.2. LightGBM

A fact that cannot be ignored is that the recorder on UAVs can generally store more than 20 flight data points per second, which made us pay more attention to the performance of the diagnostic model for processing large-scale data. Microsoft proposed the LightGBM (Light Gradient Boosting Machine), an exceedingly fast gradient boosting framework, to improve the model training speed when processing large-scale data [31]. In fault diagnosis, LightGBM has been used in diagnosing shipboard medium-voltage DC power system faults [32], rotating machinery faults [33], and classic bearing faults [34].

As an improved method of GBDT, LightGBM employs Gradient-based One-Side Sampling (GOSS) and Exclusive Feature Bundling (EFB), known as the histogram-based algorithm, to reduce the cost of calculating the gain, to speed up training, and to occupy less memory usage [35]. It was verified on UAV test flight data that the lightning LightGBM-based diagnosis model could attain a decent performance without occupying excessive computational resources [27]. As an ensemble learning model, the LightGBM still has the great potential to be employed as a meta-classifier in complicated ensemble frameworks.

### 2.3.3. MC–LGB: A Two-Step Ensemble Cost-Sensitive Model

In order to optimize the total misclassification cost, the original MetaCost framework has the defects of a decline in excessive accuracy and an increased modeling time. Based on the characteristics of the operation and maintenance data of UAVs, we attempted a modification of the original framework by changing the sampling method, improving the performance of the meta-classifier, and adjusting the organization form so as to enhance the overall performance of the diagnosis model and reduce the required computing resources while decreasing the total misdiagnosis cost.

Generating subsets for training meta-classifiers through sampling is an important part of constructing diverse meta-classifiers, which is realized by employing the bootstrap resampling in the original MetaCost framework. The sample size of different fault modes varies greatly in the actual UAV flight data, meaning that instances of specific fault modes might not be included in the subsets generated by the bootstrap method, resulting in parts of meta-classifiers lacking the ability to diagnose all existing fault modes. Therefore, stratified bootstrapping was used by us to construct the subsets, to ensure the diversity of the meta-classifiers, and to enhance the stability of the diagnosis model.

In order to achieve the expected performance, an important aspect of optimizing the ensemble learning model is improving the capacity of the meta-classifier. According to the characteristics of UAV fault diagnosis, the chosen meta-classifier needs to be lightweight enough to be integrated, while exhibiting a relatively classy diagnosis ability to avoid overfitting. In addition, the independency and diversity of the meta-classifiers are also essential in the MetaCost framework. After constructing varied subsets, different features can also be selected to obtain the diverse meta-classifiers. LightGBM, as an ensemble learning method with a decent classification performance and ultrafast training speed on actual UAV data, perfectly meets the meta-classifier requirements in the MetaCost framework. With the EFB technology, LightGBM can construct multiple decision trees under the same subset, which further enhances the diversity and independency of the meta-classifiers under the premise of stratified sampling beforehand. In addition, employing LightGBM-based models, constructed by boosting, would perhaps increase the sensitivity of the MetaCost model organized by bagging form, for processing large-scale data. In an effort to improve the global diagnostic performance while meeting the cost-sensitive function, selecting LightGBM as the meta-classifier solves the modeling speed issue with MetaCost efficiently, adapting to the limitations in computational resources. In addition, the MetaCost model with LightGBM as a meta-classifier has decent robustness in handling the grievous Gaussian noise interference [36].

Referring to the idea of the D-MetaCost algorithm proposed by Deng [37], a particular proportion of the cost-insensitive meta-classifiers before data relabeling can be integrated with the cost-sensitive meta-classifiers in the end, for offsetting some of the accuracy losses obtained after retraining. After the first ensemble, a proportion of the cost-insensitive meta-classifiers are selected based on the classification accuracy evaluation, to participate in the final ensemble.

Thanks to the above improvements, this paper proposes a two-step ensemble cost-sensitive diagnosis model based on the MetaCost framework (MC–LGB), for a UAV fault diagnosis based on operation and maintenance data. The basic workflow of the proposed model is shown in Figure 1. The MC–LGB model is divided into three stages to realize the cost-sensitive classification function: 1st ensemble stage, intermediate stage, and 2nd ensemble stage. (1) After the stratified bootstrapping, $m$ subsets from the original training set are generated and $m$ corresponding cost-insensitive LightGBM models are trained, respectively, in the 1st ensemble stage. Each LightGBM of them is integrated by $n$ decision trees. (2) In the intermediate stage, the performance evaluation procedure of the trained meta-classifiers is performed. Among the $m$ meta-classifiers, $p$ cost-insensitive LightGBMs with the best accuracy are selected to participate in the final ensemble. At the same time, based on the basic algorithm of the MetaCost framework, MC–LGB calculates the expected cost according to the cost matrix, and relabels the original training set. (3) Similar to the

1st stage, *m* subsets from the relabeled training set are generated and *m* corresponding cost-sensitive C-LightGBM models are trained, respectively. Together with the selected cost-insensitive LightGBMs, the newly trained cost-sensitive meta-classifiers are organized in the 2nd ensemble stage.

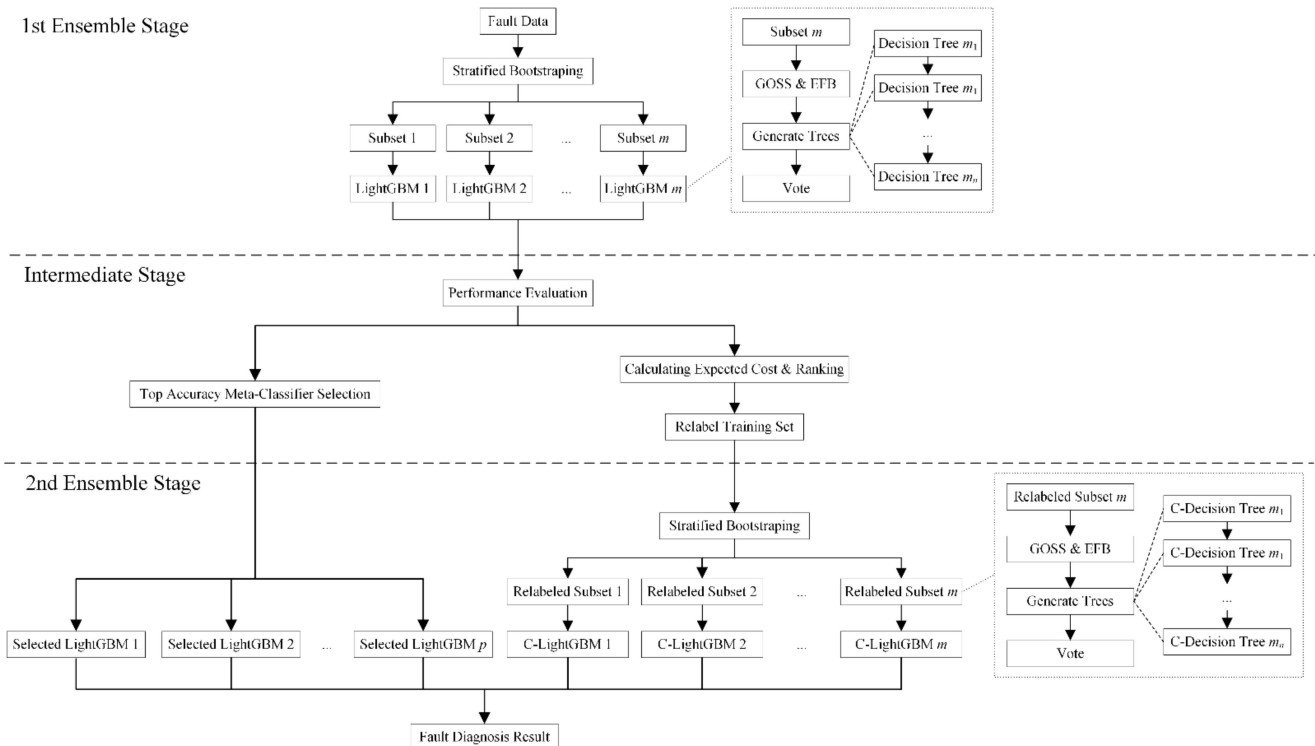

**Figure 1.** The basic workflow of the proposed two-step ensemble cost-sensitive diagnosis model (MC–LGB).

According to the workflow shown above, the main hyperparameters affecting the performance of the proposed model include the size of the subsets after the stratified bootstrapping, the hyperparameters of the generated LightGBM-based meta-classifier (such as depth limit), the proportion of selected cost-insensitive classifiers in the meta-classifier selection step, and the number of generated meta-classifiers.

## 3. Application and Experiments

### 3.1. Application Description

The operation and maintenance data are from a KPG component in the integrated navigation system of a TYW-001, a large fixed-wing UAV from BHUAS. The endurance of the TYW-001 is over 40 h with a ceiling of 8000 m, a take-off weight of 1500 kg, and an external payload of 370 kg. Apart from the fire control system and redundant backups, the flight data of the TYW-001 has a total of 457 variables, including sensor monitoring and operation input. The main functions of the KPG component include solving the orientation of the aircraft relative to the ground station; solving the information of the aircraft deviating from the predetermined course and the predetermined glide slope during microwave landing; solving the information of the aircraft deviating from the course and the glide slope during instrument landing; and providing the distance information of the runway entrance by the pointing beacon input and output signal power division, signal reception and transmission, etc.

According to the spike testing of the target KPG component, 32 relevant data features can be obtained from onboard sensors, existing 5 fault modes, and 1 normal state. Table 2 shows the distribution and corresponding criticality values of these states in our spike testing.

**Table 2.** The distribution and criticality of the different states of the KPG component.

| Fault Mode | Number of Cases | Criticality |
|---|---|---|
| Normal | 1759 | 0 |
| $F_1$ | 480 | 3.46 |
| $F_2$ | 407 | 3.31 |
| $F_3$ | 283 | 15.672 |
| $F_4$ | 113 | 23.508 |
| $F_5$ | 157 | 1.86 |

*3.2. Experimental Setup*

In order to determine whether the proposed two-step ensemble cost-sensitive diagnosis model (MC–LGB) can effectively play a role in the imbalanced KPG fault diagnosis compared with the diagnosis models employed on the entire UAV (the baseline GBDT and FCNN; the superb XGBoost; the extremely fast LightGBM; and the balanced modified CNN) [27], our experiment analyzed the performance of the proposed MC–LGB model in its overall diagnosis ability, total misdiagnosis cost optimization, and computing resource occupation. The performance of these classifiers was compared through the following metrics: accuracy, precision, recall, F1-score, MCC, AUC, and training time.

All the experiments were performed using an Intel Core i7-10700F 2.90 GHz machine with 16 GB RAM and NVIDIA GeForce GTX 3070. The codes of methods were implemented in the R-studio 1.3 and Python 3.7, with Keras 2.3.1; Scikit-learn 0.22.2.post1; Tensorflow 2.1.0; Lightgbm 2.3.1; and Xgboost 1.0.2. The hyperparameters related to the proposed model were optimized by Grid Search and manual adjustment, with the main hyperparameters of the control group being shown in Table 3. All models were evaluated in a 10-fold cross-validation.

**Table 3.** The setting of the main hyperparameters of the control group in the experiment.

| Classifier | Num. Leaves | Max. Depth | Subsample | Random State | Learning Rate |
|---|---|---|---|---|---|
| XGBoost | 64 | 7 | 0.9 | 42 | 0.001 |
| LightGBM | 56 | 6 | 0.9 | 42 | 0.001 |

| Classifier | Optimizer | Activation Function 1 | Activation Function 2 | Iterations | Learning Rate |
|---|---|---|---|---|---|
| CNN | Adam | RReLU | Softmax | 15 | 0.001 |

## 4. Results and Discussion

The aim of the experiment is to evaluate the effectiveness of the two-step ensemble cost-sensitive classifier, MC–LGB, in a multi-class UAV fault diagnosis with imbalanced data.

Adapting to the method of the cost-matrix setting, with the fault criticality given in Section 2.2, the misdiagnosis cost matrix (Table 4) for cost-sensitive classification was computed first by the fault criticality values of the different states of the KPG component.

**Table 4.** The misdiagnosed cost matrix of the different states of the KPG component.

| | | Actual State | | | | | |
|---|---|---|---|---|---|---|---|
| | | Normal | $F_1$ | $F_2$ | $F_3$ | $F_4$ | $F_5$ |
| | Normal | 0 | 3.62 | 3.46 | 16.39 | 24.58 | 1.95 |
| | $F_1$ | 3.46 | 0 | 0.15 | 12.77 | 20.97 | 1.60 |
| **Predicted** | $F_2$ | 3.31 | 0.16 | 0 | 12.93 | 21.12 | 1.45 |
| **State** | $F_3$ | 15.67 | 12.21 | 12.36 | 0 | 8.19 | 13.81 |
| | $F_4$ | 23.51 | 20.05 | 20.20 | 7.84 | 0 | 21.65 |
| | $F_5$ | 1.86 | 1.67 | 1.52 | 14.44 | 22.64 | 0 |

The calculated misdiagnosis cost matrix above was used to construct the proposed two-step cost-sensitive ensemble diagnosis model (MC–LGB). The details of each metric of the proposed method compared with that of the control group of diagnosis models and verified by actual UAV fault data, are shown in Table 5.

**Table 5.** Accuracy, precision, recall, F1-score, MCC, AUC, and training time of classifiers on the KPG dataset.

| Classifier | Accuracy | Precision | Recall | F1-Score | AUC | Total Cost | Training Time (s) |
|---|---|---|---|---|---|---|---|
| GBDT | 0.8437506 | 0.84375410 | 0.655405843 | 0.70726071 | 0.84783751 | 1747.55741 | 4.0573371 |
| XGBoost | **0.8584342** | 0.83591799 | **0.67201018** | **0.72292679** | **0.85433047** | 1733.41393 | 6.0064519 |
| LightGBM | 0.8406285 | **0.86094413** | 0.65656522 | 0.71646056 | 0.84902602 | 1767.96444 | **1.7005878** |
| FCNN | 0.8375030 | 0.79448272 | 0.64286232 | 0.68237482 | 0.79586347 | 1792.10667 | 6.3989562 |
| CNN | 0.8390625 | 0.84398075 | 0.65459342 | 0.70687547 | 0.80700821 | 1712.54131 | 9.9070521 |
| MC–LGB | 0.8208274 | 0.84215531 | 0.65531832 | 0.70772384 | 0.84405759 | **1591.51146** | 6.2960547 |

In terms of accuracy and precision (with the exception of the FCNN model's precision, which was below 0.8), all methods achieved a score that was greater than 0.82. Figure 2a–d show the comparison between the traditional classification performance metrics on the KPG dataset. It can be seen that XGBoost is the model with the strongest overall diagnostic ability on the KPG dataset (it has the highest accuracy, recall, and F1-score), with negligible disparities among the classifiers. Surprisingly, LightGBM, with insufficient diagnosis ability on the entire UAV, exhibited an excellent performance on the KPG dataset, especially in terms of precision. On these metrics, the proposed MC–LGB model showed no significant loss, which proves that our effort to enhance the overall diagnosis ability of the MetaCost framework was effective.

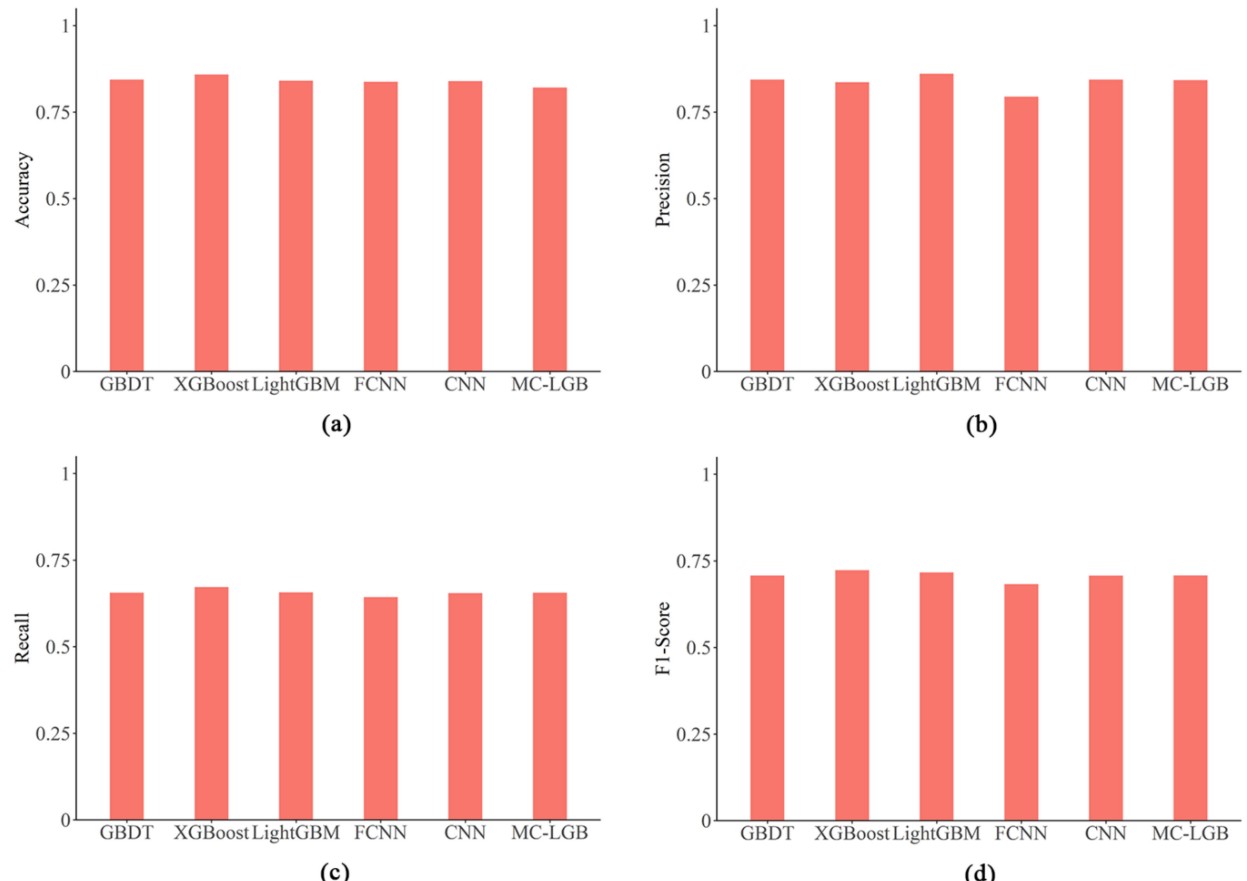

**Figure 2.** (**a**) Accuracy of classifiers on the KPG dataset; (**b**) precision of classifiers on the KPG dataset; (**c**) recall of classifiers on the KPG dataset; and (**d**) F1-Score of classifiers on the KPG dataset.

To some extent, the AUC result exhibits the responsiveness of each diagnosis model to imbalanced data, which can be visualized from the Receiver Operating Characteristic (ROC) curves in Figure 3a,b. This explains that the proposed MC–LGB model can pay attention to all operation states without showing preference for any fault mode and keep up with the best XGBoost model.

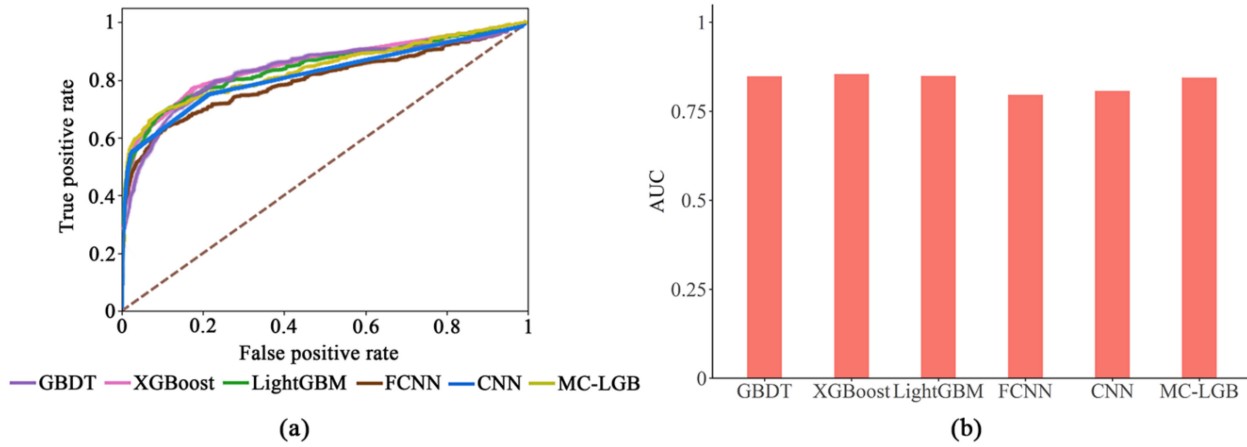

**Figure 3.** (**a**) ROC curves of classifiers on the KPG dataset and (**b**) AUC of classifiers on the KPG dataset.

For the training time of the models, shown in Figure 4a, the proposed MC–LGB model does not exceed other models in the occupation of computing resources by employing the lighting LightGBM as meta-classifiers, which greatly improves the diagnosis availability of the MetaCost-based model.

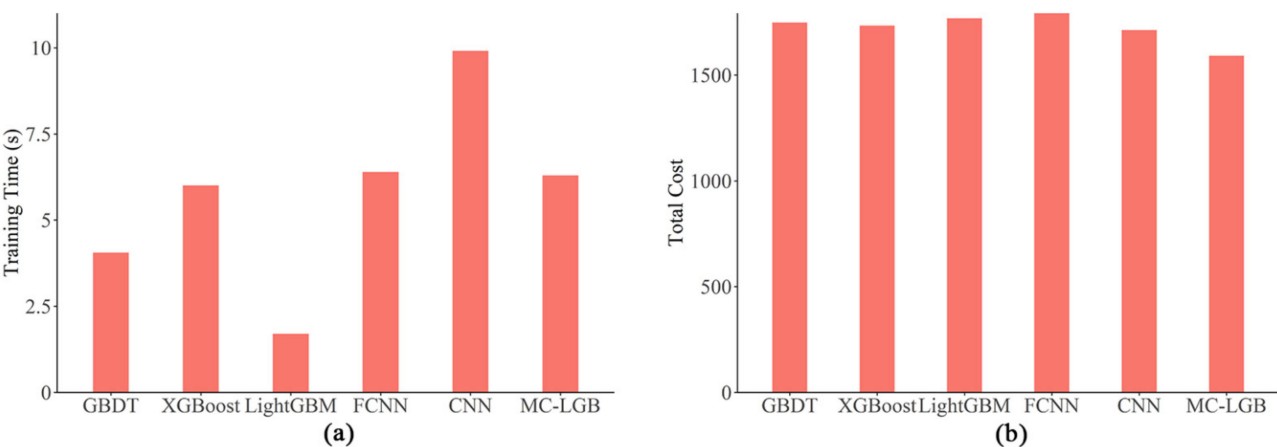

**Figure 4.** (**a**) Training time of classifiers on the KPG dataset and (**b**) total cost of classifiers on the KPG dataset.

Due to the serious imbalance of the UAV operation data and the differences among fault modes, we paid more attention to the total misdiagnosis cost, which reflects the adverse effect of model biasing in some classes. Figure 4b shows the total cost of the classifiers on the KPG dataset. In the present study, we demonstrated that the proposed MC–LGB model can significantly reduce the total misdiagnosis cost from over 1700 to 1592, even at a similar accuracy level. By preferentially improving the diagnosis ability of high criticality fault modes, the diagnosis model can reduce the cost of inevitable misdiagnoses, under the limitation of existing classification abilities.

In brief, the proposed two-step ensemble cost-sensitive method (MC–LGB) effectively complements the shortcomings of the original MetaCost framework. The MC–LGB model

can play an essential role in the future when dealing with a demand for more complex UAV fault diagnosis.

## 5. Conclusions

In order to improve data-driven fault diagnosis models within the limited computing resources, a two-step ensemble cost-sensitive diagnosis method based on the operation and maintenance data of UAV is proposed. Focusing on the overall utility of UAV diagnosis and maintenance, we managed to reduce the overall misdiagnosis cost of the diagnosis model as the core target, setting the misdiagnosis cost matrix by the fault criticality of FMECA, to guide the correction of the data-driven diagnosis model. Based on our two-step ensemble cost-sensitive method (MC–LGB), the capability of the diagnosis model can be enhanced significantly to support the ground maintenance decisions.

The experimental results based on the KPG component data of a large fixed-wing UAV show that the proposed cost-sensitive model can effectively reduce the total cost caused by misdiagnosis, without putting forward excessive requirements for computing equipment under the condition of ensuring a certain overall diagnosis performance. More specifically, compared with the other fault diagnosis models applied to the TYW-001 UAV, it is demonstrated that the proposed MC–LGB model can significantly reduce the total misdiagnosis cost from over 1700 to 1592, with an accuracy loss of only 3%. Focusing on the computing resources occupancy, the MC–LGB model also reached the average level of training time, in spite of the two-step ensemble processing.

In essence, our method improves the utility of UAV diagnosis and maintenance by restrainedly sacrificing the global accuracy of fault diagnosis. It would inevitably follow that more "non-serious faults" can be misdiagnosed, as other faults or a normal state, with this method. Once the judgment of fault influence is incorrect, it can produce the fault loss more than expected so it is necessary to emphasize the reliability of the used fault criticality values in a cost matrix setting.

It is feasible for the proposed two-step cost-sensitive model applying to actual UAV fault diagnosis. In future work, adding the ground maintenance cost and mission delay cost, caused by the misdiagnosis to the cost matrix, would be a practicable and valuable direction to explore. Since the reliable fault criticality values can be difficult to obtain in some scenarios, using the Risk Priority Number (RPN) instead of the fault criticality can also set the misdiagnosis cost matrix using our method, which requires further research on the exact RPN calculation. For the requirements of real-time fault diagnosis on UAVs in the future, the cost-sensitive diagnosis model based on the MetaCost framework cannot yet handle the data stream. Developing the cost-sensitive flight data stream mining method can be an interesting approach for fault diagnosis and the health management of UAVs.

**Author Contributions:** Conceptualization, K.Z.; methodology, K.Z. and L.Y.; software, K.Z. and C.L.; validation, K.Z. and L.Y.; formal analysis, K.Z. and C.L.; investigation, K.Z. and L.Y.; resources, G.J. and C.L.; data curation, K.Z.; writing—original draft preparation, K.Z.; writing—review and editing, K.Z.; visualization, L.Y.; supervision, G.J.; project administration, G.J. and C.L.; funding acquisition, G.J. All authors have read and agreed to the published version of the manuscript.

**Funding:** This research was funded by the Technical Research Foundation of China (Grant No. JSZL2016601A004) and Natural Science Foundation of China (Grant No. 71772010).

**Institutional Review Board Statement:** Not applicable.

**Informed Consent Statement:** Not applicable.

**Data Availability Statement:** Restrictions apply to the availability of these data. The data in this research came from BHUAS Technology Co., Ltd. Please contact K.Z. (zhengke@buaa.edu.cn) to inform about the data availability.

**Acknowledgments:** All authors would like to thank the Technical Research Foundation and Natural Science Foundation of China.

**Conflicts of Interest:** The authors declare no conflict of interest.

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
