# Peer review of "A Cost-Sensitive Diagnosis Method Based on the Operation and Maintenance Data of UAV"

_applsci, doi:10.3390/app112311116_

Round 1

Reviewer 1 Report

This paper proposes a novel UAV diagnosis method with a cost-sensitive approach to reduce the total cost caused by misdiagnosis, without too high requirements for computing equipment and ensure its overall diagnosis performance. The paper gives sufficient background information of the motivation and problems to be solved, and provide useful insights to UAV operation with reducing the diagnosis coast. The "MetaCost Framework" is applied to analyzing real UAV operation and maintenance data, and shows interesting performance compared with other classifiers. In this sense, I think this paper is worth publishing. 

A couple of points points that the paper can be improved are to provide insights in applying this method to the other UAVs. In the experiments, the authors use specific data of a large fixed-wing UAV. I am wondering what if this proposed methods are applied to the other UAV operation data, and if the results provide sufficient performance. Are there additional critical characteristics of UAV flight data other than ones shown in the section 2.1 ? Is there a application limit for this approach ? Is there a potential threat when utilizing this method in UAV industries ?

Minor points ;

p3. l. 103
"Freamwork" should be a typo.

p.4 l. 168
How were the parameters of S=100, m=0.5  chosen ? Is this specific only for this case ?

Author Response

Please refer to the attachment for specific response.

Reviewer 2 Report

My main concern regarding this paper is that the way the authors write in English is very difficult to understand. 
I strongly suggest the use of a professional service in English technical writing. 
If the paper is not well written, the readers can not get the message (even if the paper has important contributions). 

It would be interesting if the authors could give more information on the issue of imbalanced data. Also to give more information on cost-sensitive and cost-insensitive classifiers.

Some acronyms have not been defined the first time they appear:
-MCAS (line 46);
-LightGBM (line 97);
-KPG (line 105);

The conclusion of the paper is a word-by-word copy of the whole abstract. The only addition to the conclusion are the lines 381 to 383. I am not sure if this is an issue, however, usually at the abstract it should be presented a preview on what the readers are about to learn; and the conclusion should present a critical discussion about the findings of the paper.

Author Response

(The authors gave the same response as above.)

Reviewer 3 Report

In this paper, a cost-sensitive diagnosis method, based on metacost framework, is proposed, which is utilized to cope with the data of UAV possessing large-scale, high-dimensional, multi-class, etc. It is a meaningful and interesting topic. There are some confused questions that need further explanation to make this paper easy to follow. Questions are delivered as follows:

  1. In your designed metacost framework, does it support fault diagnosis in dynamic scenarios, for instance: new kind of fault? Furthermore, the idea of your deigned metacost framework is too brief, a further explanation should be offered.
  2. With respect to the priori probability in Bayesian theory, is it coming from your experience? The more details should be provided.
  3. As for the noise, some related methods are not mentioned in block diagram, as shown in Fig.1.  what method applied in this paper?
  4. The explanation of innovations of this paper are ambiguous, and some details should be provided to make your idea easy to follow, such as: diagram or others.In addition, in Fig. 1, it would be better to have separated subgraphs to deliver the idea of your two-step ensemble method.
  5. The loss design is a key point in your idea, but, I can’t find it in this paper. And, in the simulation part, the loss process figure should be provided.

In this paper, the key idea is unclear delivered, such as meta-cost frame work. However, the LightGBM, not the key innovation point, is too more information. Furthermore, the writing logic of this article is confused, and should be further modified.

Author Response

(The authors gave the same response as above.)

Round 2

Reviewer 2 Report

I would like to thank the authors for having improved the overall quality of the paper. All my concerns have been addressed and I am recommending the acceptance (after only a minor revision).

Although the English revision (specially at section 1.Introduction) has made the paper contributions more clear, I still think that section 5.Conclusions deserves a bit more attention, specially regarding the English language (it seems to me that the changes in the conclusion have not been revised). It is not a question of orthography, by the way it is written (which is a bit confusing). If possible, please, revise it.

Anyways, my recommendation is the acceptance.

Author Response

Thank you very much for your approval of our major revised version. We’ve further revised the Conslusion as follow:

“In order to improve data-driven fault diagnosis models within the limited computing resources, a two-step ensemble cost-sensitive diagnosis method based on opera-tion and maintenance data of UAV is proposed. Focusing on the overall utility of UAV diagnosis and maintenance, we managed to reduce the overall misdiagnosis cost of the diagnosis model as the core target, setting the misdiagnosis cost matrix by the fault criticality of FMECA, to guide the correction of the data-driven diagnosis model. Based on our two-step ensemble cost-sensitive method (MC-LGB), the capability of the diagnosis model could be enhanced significantly to support the ground maintenance decisions.

The experimental results based on the KPG component data of a large fixed-wing UAV show that the proposed cost-sensitive model could effectively reduce the total cost caused by misdiagnosis, without putting forward excessive requirements for computing equipment under the condition of ensuring a certain overall diagnosis performance. More specifically, compared with the other fault diagnosis models applied to the TYW-001 UAV, it is demonstrated that the proposed MC-LGB model could significantly reduce the total misdiagnosis cost from over 1700 to 1592, with only 3% ac-curacy loss. Focusing on the computing resources occupancy, the MC-LGB model also reached the average level of training time, in spite of the two-step ensemble pro-cessing.

In essence, our method improves the utility of UAV diagnosis and maintenance by restrainedly sacrificing the global accuracy of fault diagnosis. It would inevitably follow that more "non-serious faults" could be misdiagnosed as other faults or normal state with this method. Once the judgment of fault influence is wrong, it could produce the fault loss more than expected so it is necessary to emphasize the reliability of the used fault criticality values in a cost matrix setting.

It is feasible for the proposed two-step cost-sensitive model applying to actual UAV fault diagnosis. In future work, adding the ground maintenance cost and mission delay cost caused by misdiagnosis to the cost matrix, would be a practicable and valuable direction to explore. Since the reliable fault criticality values might be hard to obtain in some scenarios, using the Risk Priority Number (RPN) instead of the fault criticality could also set the misdiagnosis cost matrix by our method, which requires further research on the exact RPN calculation. For the requirements of real-time fault diagnosis on UAV in the future, the cost-sensitive diagnosis model based on the Meta-Cost framework cannot handle the data stream yet. Developing the cost-sensitive flight data stream mining method might be an interesting approach for fault diagnosis and health management of UAVs.”

Reviewer 3 Report

My suggestion is Accept.

Author Response

(The authors gave the same response as above.)
